# Herbs as a Source for the Treatment of Polycystic Ovarian Syndrome: A Systematic Review

**DOI:** 10.3390/biotech12010004

**Published:** 2023-01-03

**Authors:** Jada Naga Lakshmi, Ankem Narendra Babu, S. S. Mani Kiran, Lakshmi Prasanthi Nori, Nageeb Hassan, Akram Ashames, Richie R. Bhandare, Afzal B. Shaik

**Affiliations:** 1Department of Pharmacology, Vignan Pharmacy College, Jawaharlal Nehru Technological University, Vadlamudi 522213, Andhra Pradesh, India; 2Department of Pharmacology, Sir C.R. Reddy College of Pharmaceutical Sciences, Andhra University, Eluru 534007, Andhra Pradesh, India; 3Department of Pharmacognosy, Vignan Pharmacy College, Jawaharlal Nehru Technological University, Vadlamudi 522213, Andhra Pradesh, India; 4Department of Pharmaceutics, Shri Vishnu College of Pharmacy, Andhra University, Bhimavaram 534202, Andhra Pradesh, India; 5Department of Clinical Sciences, College of Pharmacy & Health Science, Ajman University, Ajman P.O. Box 346, United Arab Emirates; 6Center of Medical and Bio-Allied Health Sciences Research, Ajman University, Ajman P.O. Box 346, United Arab Emirates; 7Department of Pharmaceutical Sciences, College of Pharmacy & Health Science, Ajman University, Ajman P.O. Box 346, United Arab Emirates; 8St. Mary’s College of Pharmacy, St. Mary’s Group of Institutions Guntur, Affiliated to Jawaharlal Nehru Technological University Kakinada, Chebrolu, Guntur 522212, Andhra Pradesh, India

**Keywords:** herbal medicine, infertility, insulin and obesity

## Abstract

Background: Polycystic ovarian syndrome (PCOS) is a neuroendocrine metabolic disorder characterized by an irregular menstrual cycle. Treatment for PCOS using synthetic drugs is effective. However, PCOS patients are attracted towards natural remedies due to the effective therapeutic outcomes with natural drugs and the limitations of allopathic medicines. In view of the significance of herbal remedies, herein, we discuss the role of different herbs in PCOS. Methods: By referring to the Scopus, PubMed, Google Scholar, Crossref and Hinari databases, a thorough literature search was conducted and data mining was performed pertaining to the effectiveness of herbal remedies against PCOS. Results: In this review, we discuss the significance of herbal remedies in the treatment of PCOS, and the chemical composition, mechanism of action and therapeutic application of selected herbal drugs against PCOS. Conclusions: The present review will be an excellent resource for researchers working on understanding the role of herbal medicine in PCOS.

## 1. Introduction

Polycystic ovarian syndrome (PCOS) is a compounded disorder characterized by elevated androgen levels, menstrual irregularities and cysts on either one or both ovaries [1]. The difference between a normal and polycystic ovary is presented in Figure 1. PCOS is believed to be a genetically complex endocrine disorder of undetermined etiology with a complicated pathophysiology [2]. The World Health Organization (WHO) stated that PCOS affected over 116 million women worldwide in 2012. One in five Indian women are affected by PCOS [3]. Globally, 1.55 million incident cases of PCOS in women of reproductive age (15–49 years) were reported, representing an increase in the rate of 4.47% (2.86–6.37%) from 2007 to 2017 [4]. A large-scale survey conducted across India in 2020 showed that around 16% of female respondents between the ages of 20 and 29 years suffered from PCOS [5].

The precise etiology and pathophysiology of PCOS are currently being studied. Based on genetics, metabolic factors, and the clinical characteristics of PCOS, several ideas have been proposed, both in utero and postnatally. Obesity exacerbates all aspects of PCOS due to underlying metabolic disturbance. Medical treatment became the preferred treatment over surgical resection of the ovaries when options such as clomiphene and follicle-stimulating hormone (FSH) became available. There was renewed interest in the surgical treatment of PCOS when laparoscopic treatment became popular. Newer technologies such as ultrasound to image ovaries were a breakthrough in the history of PCOS [6]. In around 75% of instances, the cause of infertility is a lack of ovulation. PCOS is the primary cause of infertility in persons who have it.

### 1.1. Etiology

The major etiology behind PCOS is primary disordered gonadotropin secretions, ovarian and adrenal hyperandrogenism and disorder of insulin resistance [7]. The regulation of gonadotropin-releasing hormone (GnRH) is uncontrolled, which may lead to increased luteinizing hormone (LH) and decreased FSH; this may lead to the suppression of the response of ovarian follicles to FSH, elevated anti-Mullerian hormone (AMH), follicular arrest and the increased secretion of testosterone, estradiol and dehydroepiandrosterone [8]. Disrupted ovarian synthesis of steroid hormones in these diseases may result in an increase in circulating androgens, which may be more pronounced in women with polycystic ovarian syndrome [9] (Figure 2). Hyperinsulinism and hypogonadism are considered as the capability of insulin to stimulate gonadal and adrenal androgen production, and this hyperinsulinism is also one of the major risk factors of PCOS [10]. In PCOS, immature follicle development was observed due to increased LH levels and decreasing levels of FSH. Similarly, the increased production of androgens and reduced blood levels of aromatase were observed. Excessive androgens in PCOS is due to elevated abdominal fat, and this may lead to hyperinsulinemia and dyslipidemia. An increase in cell androgen production and hyperinsulinemia reduces sex hormone binding globulin (SHBG) to increase circulating testosterone levels. All these factors may aggravate the disease’s progression [11].

### 1.2. Pathophysiology

The interplay between neuroendocrine levels, metabolic levels and ovarian anomalies may contribute to the development of polycystic ovarian syndrome. However, insulin resistance and obesity are the leading causes of PCOS in the majority of people. Elevated insulin levels cause abnormalities in the hypothalamic–pituitary–ovarian axis [12]. This deviation may lead to the suppression of insulin’s effect on post receptors, elevation of blood levels of free fatty acids, androgens and cytokinins such as TNF-α and IL-6 and the maximum deposition of leptin and resistin in adipocytes in the abdomen [13]. Elevated androgen levels in the blood may cause adipocytes to inhibit adiponectin, resulting in an insulin sensitivity reduction as well as an insulin increase in the blood. However, insulin may also stimulate the synthesis of aldo-keto reductase 1c3 (AKR1C3), which in turn promotes the release of adipose androgens present in females [14]. Oxidative stress also has a role in the etiology of many reproductive problems and anomalies in certain women, including infertility, repeated abortions and preeclampsia. Reductions in blood amino acid levels were found to be significantly induced in PCOS women as compared to healthy controls [15]. In addition to amino acids, a deficiency in vitamins may also induce the metabolic abnormalities that might lead to the development of cysts in the ovary [16]. In a few cases, genome variation in susceptible genes such as cytochrome P1A1, CYP17A1, (CYP1A1), CYP11A, CYP19, 17β-hydroxysteroid dehydrogenase (HSD17B6), androgen receptor (AR), insulin receptor (INSR), sex hormone-binding globulin (SHBG), insulin receptor substrate 1 (IRS1) and peroxisome proliferator-activated receptor gamma (PPAR-γ) [17] is responsible for PCOS. The major signs of PCOS are abnormal blood androgen levels and sugar levels, the presence of cysts in the ovaries and premenstrual syndrome. Obesity, depression, anxiety, eating disorders and chronic metabolic syndrome are also prevalent symptoms in PCOS patients [18]. Furthermore, it will cause an increase in blood pressure and cholesterol levels, which can lead to the development of heart disease and diabetes mellitus. The diagnosis of polycystic ovarian syndrome is performed as per NICHD, AES and RDC [19]. Regular measurement of androgen levels, ultrasonography, the morphology of polycystic ovaries and the wide heterogeneity of polycystic ovarian syndrome are the currently used parameters to predict PCOS [20].

A variety of methods are used in treating PCOS, including laparoscopy and the utilization of allopathic medicines. For instance, in laparoscopy, cysts are removed from the ovaries through a surgical procedure [21,22,23]. Drugs including nafarelin, troglitazone, clomiphene, metformin and spironolactone are currently employed for the treatment of PCOS. However, these drugs can cause major complications with long-term usage, including menstrual abnormalities, nausea, vomiting and gastro-intestinal disturbances, weight gain, increased insulin resistance, less compliance, poor efficacy and more contraindications [7]. Obese people, women of childbearing age, lactating mothers and patients with cardiovascular complications [8] are at a greater risk from the usage of these drugs, due to their side effects. Hence, there is a pressing need to identify and develop drugs of plant origin, which are much more effective than existing allopathic drugs. Recently, the use of herbal medicines by healthcare professionals to treat PCOS has become a major turning point [5]. Herbal medicines are extracts of entire plants or any part of a plant that shows a major therapeutic effect, with fewer side effects when compared to conventional therapy [24]. They have a significant role in prevention, cure and rehabilitation. Herbal drugs are complex interventions with the potential for synergistic and antagonistic interactions between compounds [25]. They are essential for the management of PCOS and have fewer side effects than allopathic medicines [26]. The regular usage of herbs is safe and more efficacious in treating PCOS and suppressing the events that contribute to the development of cysts in PCOS [27]. Currently, herbal remedies are playing a prominent role in treating various chronic disorders, including PCOS. The use of herbal medicines and modifications to the diet may help in treating PCOS more effectively [8]. Different herbs will exert their activity against PCOS through a variety of mechanisms, including the suppression of prolactin levels, anti-androgenic activity, promoting follicle stimulating hormone (FSH), decreasing luteinizing hormone (LH), the induction of ovulation and restoration of glucose sensitivity, estrus cyclicity and enzyme activity (Figure 3).

Among various herbs, *Glycyrrhiza glabra* (liquorice), *Linum* (chaste berry), *Vitex negundo* (Chinese chaste tree), *Foeniculum vulgare* (fennel) and *Curcuma longa* (Turmeric) are potential plant sources that have shown effective action against PCOS [8]. 

## 2. Materials and Methods

In view of the significance of herbal remedies in the treatment of PCOS, herein, we discuss the chemical constituents, mechanisms of action and therapeutic application of selected herbal drugs against PCOS by referring to the Scopus, PubMed, Google Scholar, Crossref and Hinari databases. The following is a detailed discussion of the selected herbal drugs that are effective against PCOS via various mechanisms. 

## 3. Results

### 3.1. Herbs That Increase Ovulatory Cycles

Changes in prolactin levels and hormonal imbalances will have a significant impact on ovulatory cycles. Decreasing prolactin levels or improving the hormonal balance have a positive impact on ovulatory cycles and the treatment of PCOS. These two activities have the potential to reduce cyst formation, dissolve cysts and improve ovulatory cycles. Vitex and turmeric are two herbs that show a beneficial effect in PCOS by increasing the ovulatory cycles.

#### 3.1.1. *Vitex agnus castus*

*Vitex agnus castus* is a *Verbenaceae* family member that is also known as chaste berry, and it has been used in herbal medicine for the past 2000 years. It is a large shrub native to Europe and is also broadly disseminated in southern regions of the United States. The imbalance of estrogen levels is marked by menstrual cycle abnormalities and premenstrual syndrome, such as insufficiency of the corpus luteum and cyclical mastalgia and post-menopausal hot flashes [28,29,30]. The Vitex fruits mostly contain monoterpenoids, including bornyl acetate, limonene, 1,8–cineol, α-pinene and β-pinene [31,32], and labdane-type diterpenoids, such as viteagnusin, vitexilactone, rotundifuran and vitex lactam A. Moderate amounts of flavanoids such as luteolin, apigenin, 3-methylkaempferol, casticin, chrysoplenetin and chrysosplenol D [33,34] and iridoids such as cynaroside were also reported (Figure 4) [35,36,37]. The diterpenoid viteagnusin and flavonoids including apigenin, 3-methylkaempferol, luteolin and casticin play a major role in suppressing the prolactin levels by inhibiting dopamine-2 (D-2) receptors and increasing the ovulatory cycles [38]. Additionally, the estrogen receptor-β (ERβ)-selective action of the flavonoid apigenin will also help in increasing the ovulatory cycles [39]. All the above mechanisms can further facilitate the suppression of cyst formation in the ovary and may be useful for the treatment of PCOS.

#### 3.1.2. *Curcuma longa*

*Curcuma longa* belongs to the family *Zingeberaceae* and its rhizome is routinely used as a spice in the Asian continent. It is commonly called *Curcuma domestica Loir*, *Curcuma domestica vale*, curcuma, yellow ginger, amomum curcuma Jacq, Indian saffron and yellow root. A detailed description of the chemical constituents present in turmeric is provided elsewhere [40]. Turmeric contains a range of primary and secondary metabolites, and more than 250 phytoconstituents have been reported, including carbohydrates (60–70%), proteins (5–10%), terpenes (1.5–5%) and resins. Resins of turmeric, commonly called curcuminoids (2.5–8%), constitute the major secondary metabolites and they are responsible for the color and most of the biological properties. Curcuminoids including curcumin (diferuloylmethane) (71.50–94%), desmethoxycurcumin (6–19.4%) and bisdemethoxycurcumin (0.30–9.10%) are the three most important constituents isolated. It primarily contains phenolic compounds such as curcumins, curcuminoids, ferulic acid, eugenol, ascorbic acid, vanillic acid, caffeic acid, syringic acid, protocatechuic acid, and p-coumaric acid, and terpenoids including turmerone, α-turmerone, camphene, β-sesquiphellandrene, γ-terpinene and carotene (Figure 5) [41]. Curcuminoids have significant effects in the treatment of PCOS. They reduce the follicular sheath and improve the formation of the corpus luteum and the ovulation process. Hence, turmeric improves the histological features of polycystic ovaries. Curcuminoids also suppresses the serum levels of progesterone and elevate the levels of estradiol in women with PCOS [42]. Furthermore, their estrogenic, antihyperlipidemic, antioxidant and hypoglycemic effects are useful in managing PCOS and preventing ovarian cell dysfunction and thereby improving ovulation and fertility.

### 3.2. Herbs with Anti-Androgen Properties

Elevated blood levels of androgens are also the one of the major etiologies behind PCOS. Hence, drugs with anti-androgen activity are used in the treatment of PCOS. Herbs including *Glycyrrhiza glabra*, *Linum usitatissimum*, *Mentha spicata*, *Cocus nucifera* and *Punica granatum* have anti-androgenic action and these herbs could be useful for the management of PCOS.

#### 3.2.1. *Glycyrrhiza glabra*

*Glycyrrhiza glabra* belongs to the family Fabaceae. It contains mainly 2–9% of glycyrrhizin, glycyrrhizin acid, flavonoids, isoflavonoids, carbohydrates, amino acids and triterpenoid saponins. Liquiritigenin, isoliquiritigenin, liquiritin, isoliquiritin, glabridin and glabrene are the major phytoestrogens found in liquorice [43]. It acts as a potent anti-androgen and helps the body to maintain biosynthesis and the release of estrogen. The flavonoids of *Glycyrrhiza* (Figure 6) possess estrogenic activity and they interact with estrogenic receptors, and this results in their anti-androgenic effect. Additionally, flavonoids can help in the secretion of insulin, which reduces blood sugar levels and contributes positively to the treatment of PCOS [44]. In addition to its benefits in PCOS, liquorice has other therapeutic roles, such as cough suppression, antibacterial and antiviral activity and treatment for digestive issues, hepatitis and mouth ulcers [3].

#### 3.2.2. *Linum usitatissimum*

*Linum usitatissimum* is commonly called linseed and it belongs to *Linaceae*. It contains 30–40% of fixed oils, 6–10% mucilage, 25% proteins, saturated and unsaturated fatty acids and lignans (Figure 7). Major lignans include secoisolariciresinol and secoisolariciresinol diglycoside-SDG, and the mucilage of fiber is rich in *l*-galactose, *d*-xylose, *d*-galacturonic acid and *l*-rhamnose. The secondary metabolites of *Linum* regulate estrogen production in the body and promote fertility rate as well as the menstrual cycle [45]. Similarly, they decrease androgen levels and considerably reduce elevated levels of testosterone in the blood, which is useful for treating PCOS [46]. Flaxseed also reduces symptoms associated with PCOS, such as hyperandrogenism and hirsutism. Preclinical studies confirmed that the supplementation of flaxseed in the diet decreased the androgen levels in female rats, improved the formation of the corpus luteum and reduced the number of cysts in ovarian follicles [47]. Hence, the regular intake of flaxseed through dietary supplementation could significantly suppress the ovarian volume and follicle size in the ovaries and regulate the frequency of menstrual cycles. It is also used in various disorders, such as neoplasms, diabetes mellitus, arrhythmia, obesity and clotting problems in blood vessels [3].

#### 3.2.3. *Mentha spicata*

*Mentha spicata* is a Labiatae member that has been used in culinary applications for many years. It contains carotenoids, such as lutein, and flavonoids and their analogs, including catechin, rutin, xanthomicrol, quercetin-4-glucoside, 5,6-dihydroxy-7,8,3,4-tetramethoxyflavone, sorbifolin, thymosin, hesperidin, gallocatechin-gallate, thymonin, sideritoflavone, ladanein, narirutin, luteolin -7-O-rutinoside, isorhoifolin, eriodictyol-7-O-glucoside and 5-O-demethylnobiletin [48], as well as phenolic compounds such as rosmarinic acid, caffeic acid, salvianolic, dehydro-salvianolic acids and cinnamic acids. The major components that possess good antioxidant effects are lutein, rutin, rosmarinic acid and caffeic acid (Figure 8) [49]. During a 30-day in vivo study of laboratory animals, these compounds reduced the free and total testosterone levels, as well as the degree of hirsutism. Spearmint improves ovarian cysts in PCOS by reducing atretic follicles and enhancing graafian follicles. It has anti-inflammatory, anti-diabetic and anti-cancer properties [50,51]. Mentha regulates the blood ratio of LH and FSH. Based on this regulation of LH/FSH in the blood, it could be useful for the treatment of PCOS. Several preclinical studies have shown that spearmint has anti-androgen properties [52]. 

#### 3.2.4. *Cocos nucifera*

*Cocos nucifera* belongs to the family *Arecaceae*. The oil contains primarily alpha-tocopherol and lauric acid, and, in the case of roots, it has rich phenolic compounds, such as flavonoids and saponins [53]. It also contains lupeol methyl ether, skimmiwallin and isoskimmiwallin (Figure 9). *C. nucifera* contains twenty-five volatile and semi-volatile phytoconstituents [54]. The flavonoid present in *Cocos* has hypoglycemic effects and reduces blood glucose levels. The lipid methyl (9*Z*,12*Z*)-9,12-octadecadienoate present in this plant possesses anti-androgenic properties [55]. *C. nucifera* regulates the blood levels of sex hormones such as FSH and LH [56]. Preclinical research has shown that *Cocos nucifera* has a beneficial effect in modifying the histology of the ovaries in PCOS patients, in terms of cyst size and number. Similarly, it also suppresses the weight of the ovary and increases the uterus weight. Based on the regulation of hormones, it could be useful for preventing cyst formation in the ovaries. In India, infusions of a coconut inflorescence taken orally are used to treat menstrual cycle disorders [57]. Similarly, upon oral administration, coconut milk has a contraceptive property [58,59,60].

#### 3.2.5. *Punica granatum*

*Punica granatum* belongs to the family *Punicaceae* and possesses large amounts of folic acid, vitamins (B2, C and B1), sugars, organic acids and pantothenic acid. Unsaturated and saturated fatty acids are commonly seen in the seeds. The phenolic compounds (catechins and flavonoids) (Figure 10) and phytosterols found in the seed extract have good effects in reducing the complications of PCOS. In women who included *Punica granatum* in their regular diet, the blood levels of free testosterone, serum estrogen and androstenedione hormone were normalized [61]. Based on the various research conducted on *Punica granatum*, it is found that the usage of *Punica granatum* reduces the complications associated with PCOS [3].

### 3.3. Herbs That Restore Glucose Sensitivity, Estrus Cyclicity and Enzyme Activity 

Decreasing insulin sensitivity and elevated blood glucose levels are also two of the major symptoms observed in women suffering from PCOS. As a result, drugs that increase insulin sensitivity are included in PCOS treatment. Herbs such as *Cinnamomum cassia* and *Aloe vera*, which have the same mechanism, can reduce blood glucose as well as regulate the estrus cycle and could be useful.

#### 3.3.1. *Cinnamomum cassia*

*Cinnamomum cassia* belongs to the family *Lauraceae*. An array of polyphenolic compounds has been reported in *Cinnamomum cassia*, including cinnamyl alcohol, linalool, eugenol, eugenol acetate, methyl eugenol and benzaldehyde, along with procyanidins such as cinnamaldehyde, cinnamyl acetate, caryophyllene, monoterpene, pinene, hydrocarbon, benzyl benzoate, phellandrene, safrole, cymene and cineol (Figure 11) [62]. Cinnamomum contains 80.59% carbohydrates, 59.55–53.1% dietary fiber, 9.5–10.5% moisture, 3.89–4.65% protein, 3.55% ash and vitamins [63]. Phenolic compounds regulate blood insulin levels, as well as promoting glucose uptake and the synthesis of glycogen in the liver [64]. Cinnamon extract improves insulin selectivity in women with PCOS. The procyanidins and polyphenols in cinnamon are responsible for the hypoglycemic effect by stimulating the insulin signaling pathway. Cinnamon is used as an adjunctive in the treatment of PCOS through oral supplementation during the luteal phase, where it could regulate progesterone levels. Similarly, taking cinnamon on a daily basis will help to normalize the menstrual cycle and effectively suppress polycystic ovary syndrome [65]. The ingredients possess good anti-inflammatory and antioxidant properties. They raise the levels of superoxide dismutase (SOD), glutathione peroxidase (GPX) and catalase (CAT) in the blood, while lowering the level of malondialdehyde (MDA) and increasing the likelihood of pregnancy [66]. Moreover, it lowers the fasting blood glucose level and insulin, total cholesterol (TC), low-density lipoprotein (LDL) and triglyceride (TG) concentrations [67]. A series of clinical trials with 183 participants were performed to compare the effects of cinnamon alone and a mixture of herbs to a placebo or a control. Since the I2 of 17% showed that there was no statistical difference between the studies (*p* = 0.30), the results were combined using a fixed effect model. The overall effect size showed that there was a statistically significant difference in the LDL levels of PCOS patients who consumed cinnamon alone and those who used a mixture of herbs. This means that PCOS patients who ate cinnamon alone or used a mixture of herbs had significantly lower LDL levels.

#### 3.3.2. *Aloe vera*

*Aloe vera* is a *Liliaceae* plant that contains secondary metabolites such as anthraquinone derivatives, flavonoids, phytosterols, polyphenols and other nutrients. Aloe emodin and barbaloin (Figure 12) are the major constituents of *Aloe vera*. It was observed that women treated with *Aloe vera* gel experienced the partial reversion of estrous cyclicity and improved steroidogenic activity. Similarly, in pre-clinical studies on rats, aloe vera not only suppressed 3*β*-HSD activity and 17*β*-HSD activity but also reduced the ovary weight, resulting in the suppression of overall androgen secretion. Additionally, it also increased estrogen synthesis by stimulating the flux of the steroidogenesis pathway. It can restore glucose sensitivity, the estrus cycle and the plasma levels of lipoproteins, besides suppressing the biogenesis of cholesterol in the liver [68]. Aloe vera also has a regulating effect on blood lipid and glucose levels, and hence this activity is useful in treating PCOS due to metabolic disturbances [69,70].

### 3.4. Herbs That Promote FSH and Decrease LH Secretions

In PCOS, a common complication is elevated levels of LH and decreased levels of FSH. Hence, drugs that have the ability to elevate the levels of FSH and reduce the concentrations of LH are beneficial in the treatment of PCOS. Herbs including *Foeniculum vulgare*, *Panax ginseng* and *Cimicifuga racemosa* have such actions; hence, they are useful for the treatment of PCOS.

#### 3.4.1. *Foeniculum vulgare*

*Foeniculum vulgare* is usually called fennel and belongs to the family *Apiaceae*. It has a volatile oil content of 4–5%. Numerous chemical constituents with many therapeutic activities have been seen in fennel. The constituents include *trans*-anethole, *α*-pinene estragole, fenchone, 1,8-cineole, *β*-carotene, myristicin, limonene, *β*-sitosterol, cinnamic acid, caffeic acid, ferulic acid, fumaric acid, benzoic acid, *p*-coumaric acid, vanillic acid, kaempferol, quercetin, rutin and vanillin. Fennel contain 50–60% of anethole, phenolic esters, 18–22% of fenchone, fixed oils and proteins [71], as well as vitamins such as *α*-tocopherol, ascorbic acid, *β*-tocopherol, *γ*-tocopherol and *δ*-tocopherol (Figure 13) [72]. Fennel vitamins have high antioxidant activity and protect the cells from oxidative damage. Anethole promotes menstruation, facilitates birth and also induces estrogenic properties in the ovarian follicle. All these may contribute to the treatment of PCOS. Other pharmacological properties of fennel are useful for the treatment of helminthic infections, neurological disorders and hirsutism. Further, it has tumor suppression, anti-diabetic and hepatoprotective properties [73,74,75].

#### 3.4.2. *Panax ginseng*

*Panax ginseng* is known as “the king of herbs” and has been used for more than 2000 years. The saponins found in ginseng are the active molecules responsible for its useful therapeutic activity, and they include major ginsenosides, namely ginsenoside Rb1, Rb2, Rc, Rd, Re, Ro and Ra, and minor ginsenosides (Figure 14). *P. ginseng* can stimulate the growth of estrogen receptor (ER)-positive (þ) cells in vitro. Ginsenoside Rb1 and Rg1 can stimulate ERs with estrogen-like activity [43]. Hence, this can significantly elevate serum estradiol while suppressing follicle-stimulating hormone (FSH) and luteinizing hormone (LH). A significant decrease in plasma LH levels may be beneficial and effective for improving the fertility rate in PCOS anovulation patients [43]. Ginseng can also help with postmenopausal symptoms such as insomnia, anxiety and depression. It is often used as a natural estrogen replacement therapy, as well as because it can modify the estrus cycle and shows significant estrogenic effects, as suggested by the reversal of atrophy of the vagina and uterus with the upregulated expression of ER*α* and ER*β* in the reproductive tissue. All the above-discussed features of ginseng are positive factors that could contribute to the treatment of PCOS [76].

#### 3.4.3. *Cimicifuga racemosa*

*Cimicifuga racemosa* is also called black cohosh and belongs to the family *Ranunculaceae*. Over the last several decades, a large number of researchers have focused nearly entirely on two groups of compounds (triterpene glycosides and phenolic acids). The major phenolic components of black cohosh include hydroxycinnamic acids, ferulic acid, isoferulic acid and caffeic acid, as well as their condensation products with glycolyl phenylpropanoids, commonly known as cimicifugic acids (Figure 15) [77]. These are responsible for the suppression of cysts in the ovary. These compounds exhibit their effects by specifically interacting with the hypothalamus and pituitary estrogen receptors (ERα). The binding of compounds to α-estrogen receptors in the pituitary gland will reduce LH production. The flavonoids decreased the blood levels of LH and also improved pregnancy rates in patients who had ever used clomiphene during a menstrual cycle [78].

#### 3.4.4. *Pimpinella anisum* L.

*Pimpinella anisum* L. is commonly called anisum or sweet cumin and belongs to the family *Apiaceae*. It contains 9% moisture, 35% sugars, 18% protein, 16% lipids, 7% ash, 5% starch, 12–25% crude fiber, as well as 2–7% essential oil [79]. Additionally, it contains oleoresin, which is a yellowish-green to orange-brown liquid. The major constituents of aniseed oil are *trans*-anethole (90%), anisketone, anisaldehyde and methyl chavicol. Other minor constituents present in anisum include *γ*-himachalene (2–4%), *trans*-pseudo isoeugenol 2-methylbutyrate (1.3%), *p*-anisaldehyde (1%) and methyl chavicol (0.9–1.5%) (Figure 16) [80,81]. Anethole helps to relieve oligomenorrhea and improve quality of life in women who are undergoing treatment for PCOS. The phenolic ingredients possess phytoestrogenic features, which may play a greater role in the regulation and improvement of menstrual cycles and LH/FSH secretion in women with PCOS and play an important role in relieving PCOS complications [74].

#### 3.4.5. *Trigonella foenum-graecum*

*Trigonella foenum-graecum* is commonly called “fenugreek” and belongs to the family *Fabaceae*. The fenugreek fruit contains 8–10% moisture, 45% carbohydrates, 15–28% protein, 6–12% lipid, 4–8% ash, 8–16% fiber and 0.2–0.3% essential oil [82]. The primary components of fenugreek seed oil responsible for the suppression of cysts are *β*-pinene, *β*-caryophyllene, camphor and neryl acetate (Figure 17). In the case of volatile oil, the fenugreek seed is rich in sesquiterpenes, n-alkanes and some oxygenated components [83]. It also contains phytosterols, terpenoids, flavonoids (naringenin, saponaretin, lilyn, kaempferol, isovitexin, orientin, vitexin, isoorientin, luteolin and quercetin) alkaloids (choline, trigonelline and carpaine) and saponins (fenugrin, foenugracin, trigonoesides, glycoside, yamogenin, smilagenin, yuccagenin, sarsasapogenin, hederagin, tigonenin, diosgenin) [84]. Clinical trials demonstrated that it was effective in alleviating PCOS symptoms when women took two capsules daily through the diet [85,86]. Anisum reduced the cyst size, as well as ovary volume, in women who received it daily as a supplement to their diet over a period of 90 days [87]. Similarly, it also decreases the LH/FSH ratio; significant maintenance of the menstrual cycle was seen following oral supplementation. Based on these actions, this herb could have a useful and significant effect on PCOS [74].

### 3.5. Effective Ovulation Induction Agents

The most common complication of PCOS is infertility or frequent pregnancy termination due to the patient’s lack of carrying capacity. As a result, drugs used to stimulate ovulation are included in PCOS treatment. Herbs such as *Zingiber officinalis* and *Tribulus terrestris*, which have the same action and promote ovulation, might be useful for the treatment of PCOS.

#### 3.5.1. *Zingiber officinalis*

*Zingiber officinalis* is commonly called ginger and belongs to the family *Zingeberaceae*. The essential oil of ginger contains around 60-65 compounds. The major active phytochemicals are geraniol, gingerol, curcumin, α-curcumene, geranial, neral, borneol, linalool, β-sesquiphellandrene, afarnesene, sabinene, camphene, *gamma*-terpinene and terpinen-4-ol. The resin component of ginger comprises the constituents paradol, zingerol, zingiberene, zingiberon, shogaol, ascorbic acid, *β*-carotene, *p*-coumaric acid and caffeic acid [88,89]. Additionally, ginger also contains flavonoids and phenolics that are beneficial in PCOS. Gingerol and shogaol are potent antioxidant compounds, along with zingerone and a mild amount of oily resin ginger; all of these have demonstrated an anti-prostaglandin effect by inhibiting prostaglandin production and suppressing arachidonic acid production (Figure 18) [90]. Ginger will increase the fertility index, the testosterone level in serum and the testes and seminal vesicle weight, and increase the motility of sperm as well as the sperm count in males. The flavonoids and phenolic compounds in ginger could maintain the balance of estrogen and progesterone, with their specific pharmacological and physiological effects. Similarly, they could regulate the sex hormones in the blood [91]. Ginger’s phytoestrogen component can balance the estrogen to progesterone ratio, and thus it could be used to treat PCOS [92]. 

#### 3.5.2. *Tribulus terrestris*

*Tribulus terrestris* is commonly known as Gokharu, or puncture vine, and belongs to the family *Zygophyllaceae*. The major chemical constituents are furostanol and spirostanol, and saponins such as tigogenin, diosgenin, gitogenin, hecogenin, neogitogenin, neohecogenin, chlorogenin, ruscogenin, protodioscin and protogracillin. Kaempferol, kaempferol-3-glucoside, kaempferol-3-rutinoside and tribuloside are also potent therapeutic compounds derived from *Tribulus* (Figure 19). Saponins obtained from TT possess hypoglycemic properties [93]. TT significantly reduced the serum glucose level, serum triglyceride level and serum cholesterol in a study. It normalized estrous cyclicity, steroidal hormonal levels and ovarian follicular growth. Many compounds from *Tribulus* are effective ovarian stimulants and act as fertility tonics for women, making it a good choice for women with polycystic ovaries [94].

## 4. Discussion

Previously, a meta-analysis of several herbs was conducted to compare active ingredients that are effective against PCOS [95,96]. After studying the chemical contents of the herbs that are useful for treating PCOS, we believe that phenylpropanoids, flavonoids and their glycosides play a significant role in treating PCOS through several mechanisms. Other plant compounds with anti-PCOS efficacy include steroids, such as steroidal saponins and cycloartane derivatives, terpenoids, phenolics, catechins, resins, lignans and curcuminoids. As a result, we believe that a polyherbal formulation including a combination of flavonoids, phenylpropanoids and other ingredients with various mechanisms of action may be useful in controlling PCOS. A number of researchers have conducted human randomized controlled clinical studies, involving a total of 361 individuals, including 184 in the experimental group and 177 in the control group. On this basis, all patients in the group received luteal support combined with traditional Chinese medicine, whereas the control group received luteal support alone. For the meta-analysis, the researchers used random effects models. The combined SMD value of the three studies was −2.38, with a 95% confidence interval of [−2.82, −1.93], and it was statistically significant (Z = 10.50, *p* < 0.00001), indicating that the integrated traditional Chinese and Western medical treatment was superior to a purely Western medical treatment in lowering serum plasminogen activator inhibitor type-1 (PAI-I) levels in PCOS patients at risk of abortion. Similarly, polyherbal formulations containing three or more herbs in a tablet, capsule or pill have a favorable impact on PCOS. After 3 months of therapy, extremely significant benefits were discovered in terms of lowering discomfort during menstruation, reducing menstrual irregularity, normalizing follicular growth and ovulation and considerably reducing obesity in PCOS patients [97]. Hence, we conclude that a polyherbal formulation will have a greater impact in terms of decreasing the symptoms as well as treating PCOS in a more efficient manner.

## 5. Conclusions

PCOS is the most common hormonal illness in women from adolescence to pre-menopause, with a variety of complications, including infertility, metabolic and cardiovascular issues and long-term health issues that can last a lifetime. Synthetic medications have shown excellent management for the treatment of PCOS, but substantial adverse drug reactions make their value for long-term cure questionable. To enhance recovery rates and acceptance, patients are increasingly relying on herbal therapy as an alternative to synthetic medications for the control and treatment of PCOS. The current review provides a thorough review of herbs that are beneficial for PCOS and related complications. We have reviewed various key medicinal herbs, their primary chemical constituents and their specialized significance in PCOS management. We are certain that our evaluation will be of significant use to researchers working on herbal therapies to treat PCOS.

## Figures and Tables

**Figure 1 biotech-12-00004-f001:**
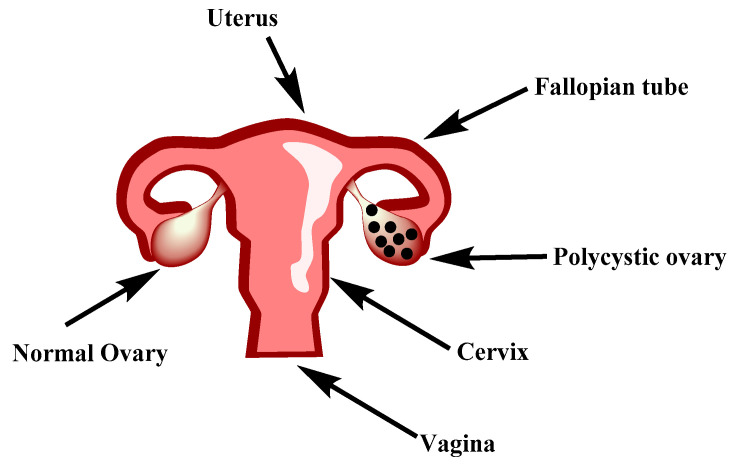
Differentiation between a normal and polycystic ovary.

**Figure 2 biotech-12-00004-f002:**
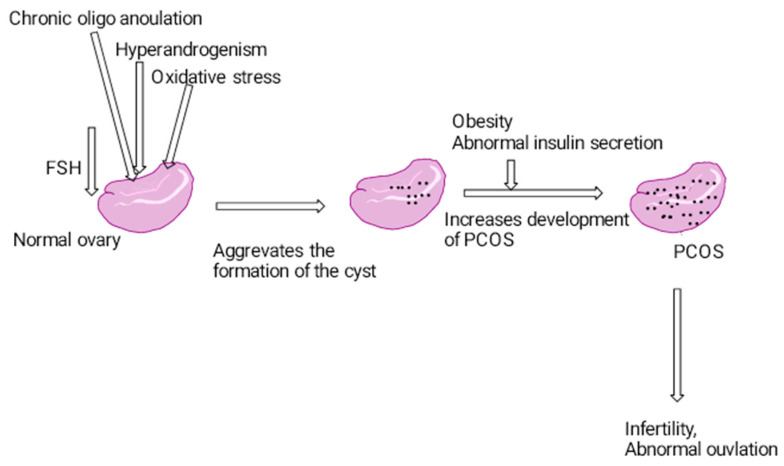
Pathogenesis of polycystic ovarian syndrome.

**Figure 3 biotech-12-00004-f003:**
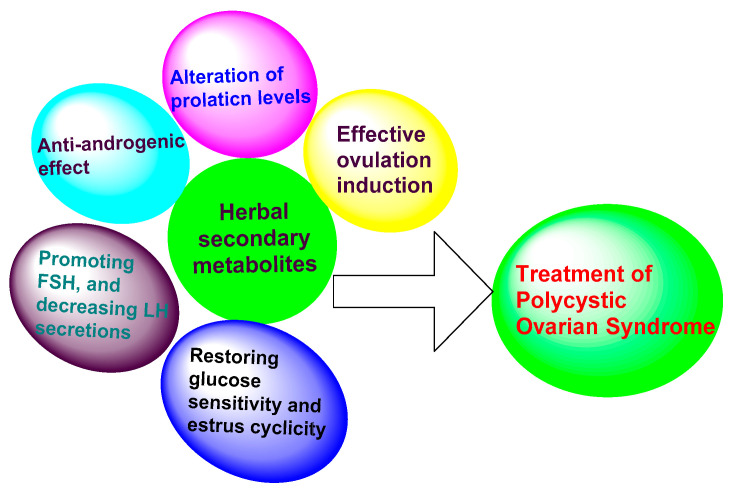
Mechanisms through which different herbal secondary metabolites are active in the treatment of PCOS.

**Figure 4 biotech-12-00004-f004:**
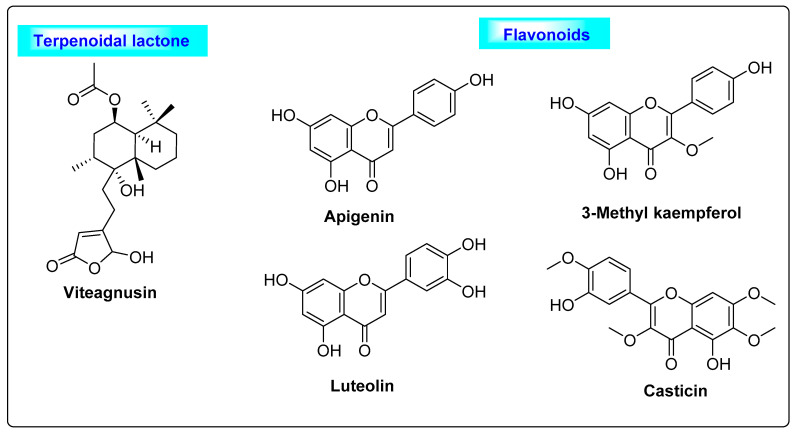
Terpenoidal lactone and flavonoids of *Vitex agnus castus* responsible for improving ovulatory cycle.

**Figure 5 biotech-12-00004-f005:**
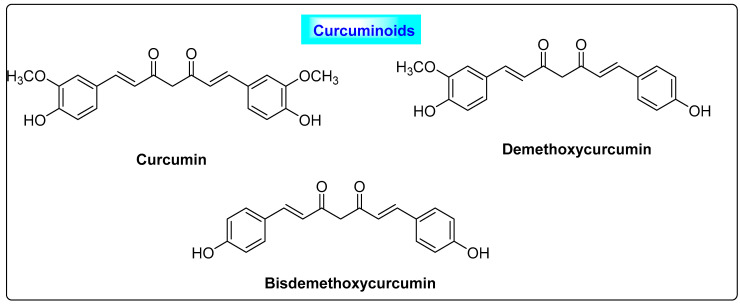
Curcuminoids seen in *Curcuma longa* responsible for improving the PCOS condition.

**Figure 6 biotech-12-00004-f006:**
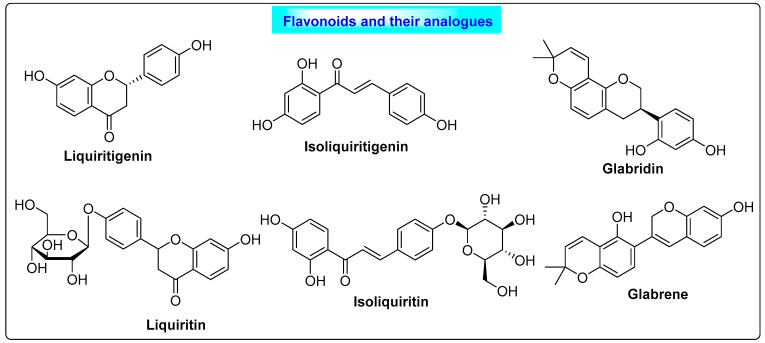
Flavonoids of liquorice that show beneficial effects in PCOS through anti-androgenic activity.

**Figure 7 biotech-12-00004-f007:**
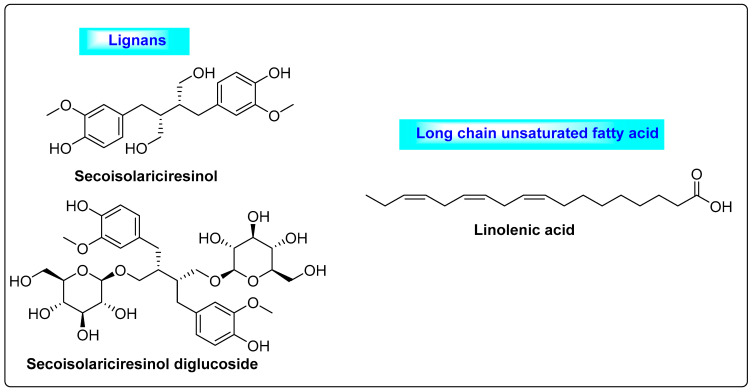
Lignans and polyunsaturated fatty acids present in *Linum usitatissimum* with anti-androgenic activity.

**Figure 8 biotech-12-00004-f008:**
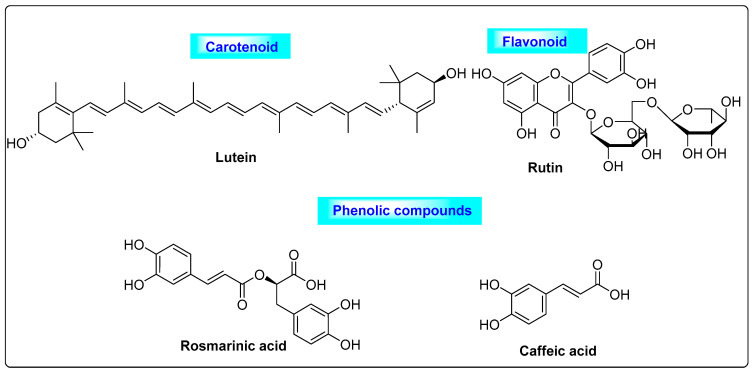
An array of different secondary metabolites of *Mentha spicata* that possess anti-androgenic activity.

**Figure 9 biotech-12-00004-f009:**
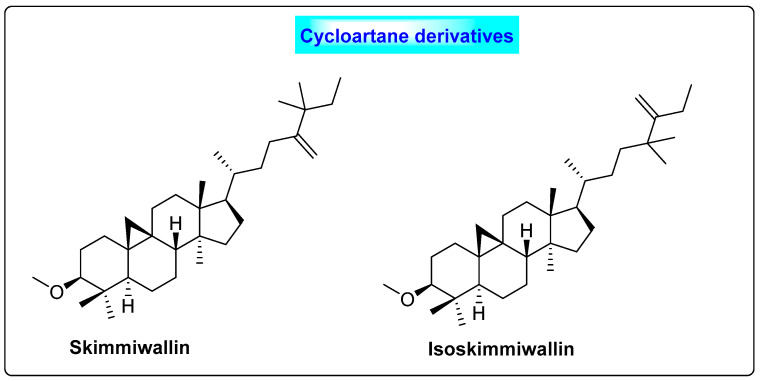
Triterpenoids of *Cocos nucifera* that possess anti-androgenic activity.

**Figure 10 biotech-12-00004-f010:**
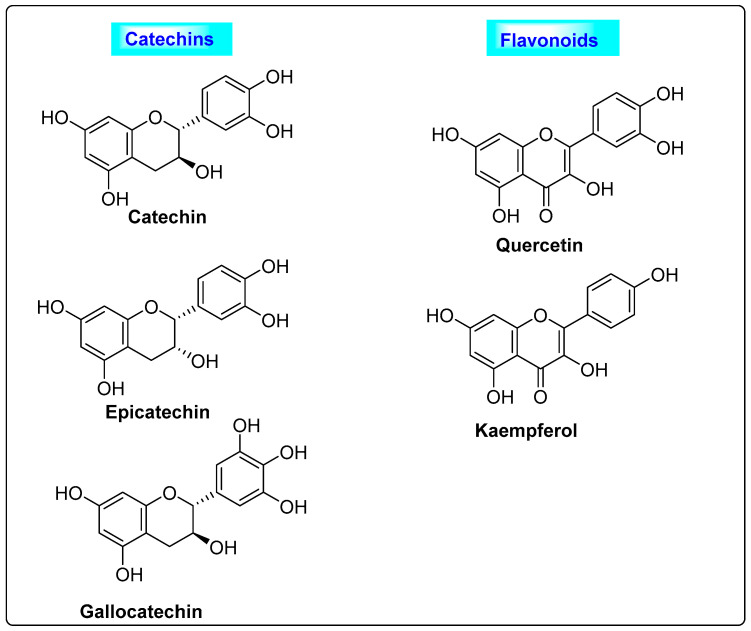
Phenolics present in Punica granatum having anti-androgen properties.

**Figure 11 biotech-12-00004-f011:**
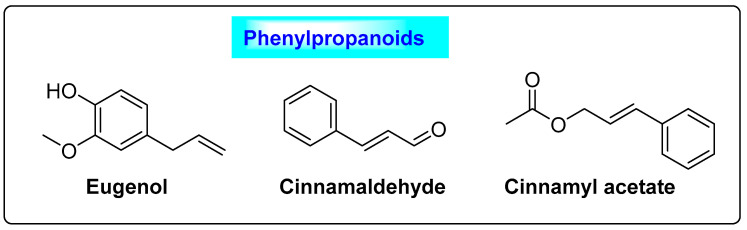
Phenylpropanoids of *Cinnamomum cassia* that exhibit enzymatic activity.

**Figure 12 biotech-12-00004-f012:**
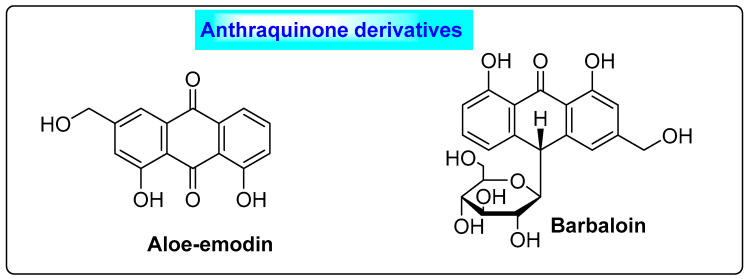
Secondary metabolites of *Aloe vera* responsible for altering enzymatic activity, which could be beneficial in PCOS.

**Figure 13 biotech-12-00004-f013:**
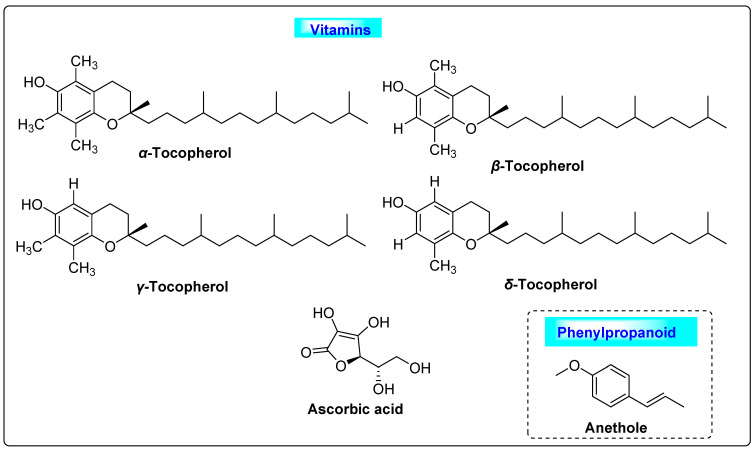
Vitamins and phenylpropanoids of *Foeniculum vulgare* that possess usefulness in PCOS.

**Figure 14 biotech-12-00004-f014:**
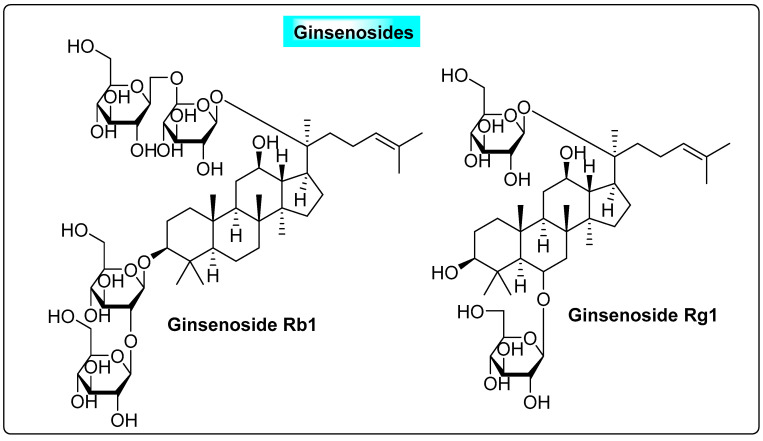
Useful anti-PCOS ginsenosides present in *Panax ginseng*.

**Figure 15 biotech-12-00004-f015:**
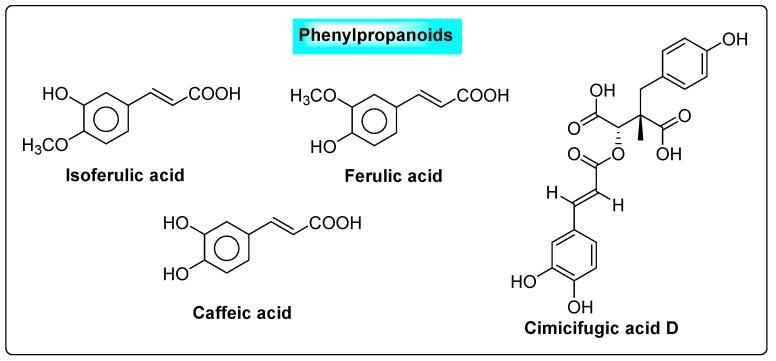
Phenylpropanoids present in *Cimicifuga racemosa*.

**Figure 16 biotech-12-00004-f016:**
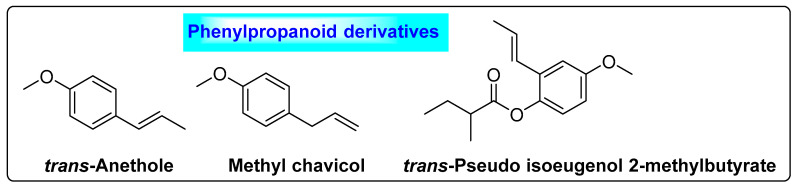
Phenylpropanoids present in *Pimpinella anisum* L. that possess anti-PCOS activity.

**Figure 17 biotech-12-00004-f017:**
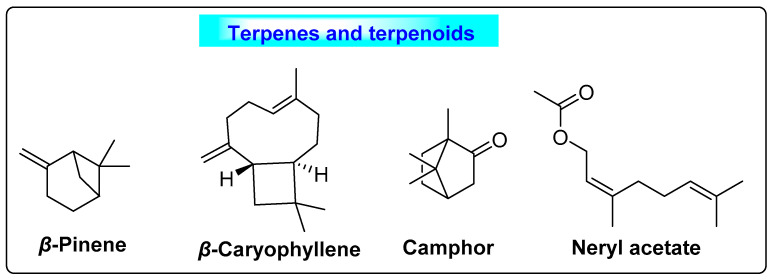
Terpenes and terpenoids of *Trigonella foenum-graecum* that suppress cyst formation.

**Figure 18 biotech-12-00004-f018:**
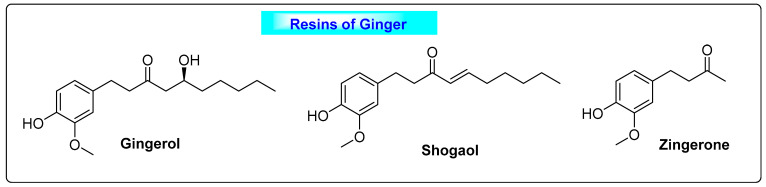
Resinous substances of *Zingiber officinalis*.

**Figure 19 biotech-12-00004-f019:**
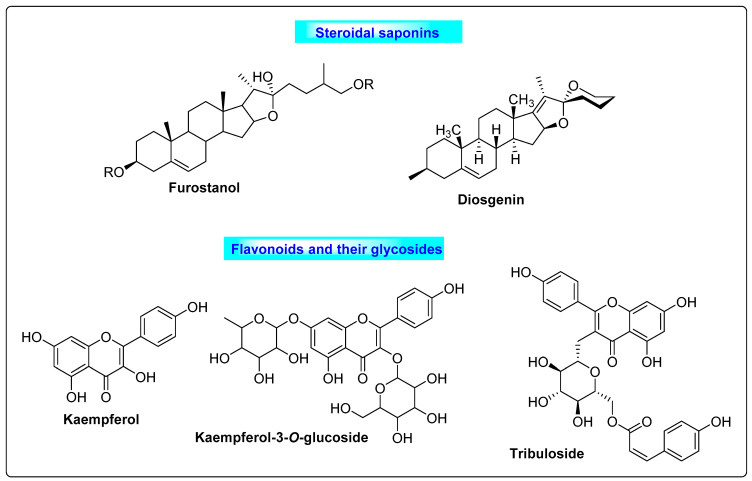
Saponins and flavonoids present in *Tribulus terrestris* that have positive effects on PCOS by inducing ovulation.

## Data Availability

Data sharing is not applicable. No new data were created or analyzed in this study.

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
