# Peer review of "Herbs as a Source for the Treatment of Polycystic Ovarian Syndrome: A Systematic Review"

_biotech, 2023, doi:10.3390/biotech12010004_

Round 1

Reviewer 1 Report

Comments for biotech-2030319

The manuscript review the significance of the herbal remedies and the role of different herbs in polycystic ovarian syndrome. The content of the manuscript is interesting, but there are some problem in experimental design should be solved before the manuscript been considered for publication.

Substantial revisions

Q1: In the past, herbs has been used for the clinical treatment of polycystic ovarian syndrome. The literature was organized and published in 2019 (DARU J Pharm Sci (2019) 27:863–877), please add a list of the used herbs that is not covered by this paper or has been newly discovered in human clinical practice, and discuss whether the medicinal functions of these herbs are antagonistic to other western medicines and take precautions.

Q2:  In the past, there have been related literatures discussing this similar issue with Meta-analysis, which are worthy of reference. Please add that if the statistical analysis method (Meta-analysis) of herbs is also used for the comparative analysis of active ingredients and polycystic ovarian syndrome characterization, so that readers can have more information to understand that different plants may have the same or different active ingredients for polycystic ovarian syndrome.

Ref 1: The Effect of Herbal Medicine Supplementation on Clinical and Para-clinical Outcomes in Women With PCOS: A Systematic Review and Meta-analysis  (Ainehchi et al., 2019) .

Ref 2: Review Article Meta-analysis of therapeutic efficacy and effects of integrated traditional Chinese and Western medicine on coagulation and fibrinolysis system in patients with threatened abortion and polycystic ovary syndrome (Am J Transl Res 2022;14(5):2768-2778).

Q3: Most of the studies have used single herbs for the treatment of polycystic ovary syndrome, please add a discussion on the related studies of compound herbs in the treatment of polycystic ovary syndrome.

Reviewer 2 Report

The English grammer and spelling needs to be rechecked 

Figures need to be improved i.e. Fig 1 the arrows should point to the struture not to the writing . Fig 2 Needs to be revised with different arrows and colours for the diffeernt mechanisms 3.1. 3.2 etc as in the paper, and structures ovary etc

Using Scopus, Pubmed...... is good but does this ensure that all relevant information has been included?

Structures of the  compounds e.g. fig 3, 4... is nice but do different colours have a significance?

References are OK but the grammer style is not consistent for all.

Round 2

Reviewer 1 Report

Comments for biotech-2030319

The manuscript review the significance of the herbal remedies and the role of different herbs in polycystic ovarian syndrome. In the revised manuscript, my questions in experimental design has been responded by authors. Thus I suggest that the manuscript be considered for publication.

Reviewer 2 Report

1. Although much improved the English still needs to be edited in the paper.

Example - Polycystic ovarian syndrome (PCOS) is a neuroendocrine metabolic disorder characterized by irregular menstrual cycle. Treatment for PCOS by using synthetic drugs is effective. However, the patients are much attracted towards natural remedies due to the limitations of allopathic medicines as well as the effective therapeutic outcome with natural drugs compounds. In view of the significance of the herbal remedies, herein we discussed the role of different herbs in PCOS. Methods: By referring to Scopus, PubMed, Google Scholar Crossref and Hinari databases, a thorough literature search was conducted and data pertaining to the effectiveness of the herbal remedies against PCOS was collected. Results: In this comprehensive review, we discussed about the selected herbal drugs  helpful against PCOS that act through different mechanisms. Ssignificance of the herbal remedies in the treatment of PCOS, the chemical composition, mechanism of action and therapeutic application of selected herbal drugs against PCOS was dealt for the ready reference of the readers working in this area of research. Conclusions: The present review will be an excellent resource for the researchers working on understanding the role of herbal medicine in the area of PCOS.

2. Whats the difference between the two figures 2?

3. The rest of the figures are now clear and concise.

4, The references seem to be in the style of the journal now.
